# Sexual orientation identity and tobacco and hazardous alcohol use: findings from a cross-sectional English population survey

Lion Shahab,[1] Jamie Brown,[1,2] Gareth Hagger-Johnson,[3] Susan Michie,[2] Joanna Semlyen,[4] Robert West,[1] Catherine Meads[5]

[1]Department of Behavioural Science and Health, University College London, London, UK
[2]Department of Clinical, Educational and Health Psychology, University College London, London, UK
[3]Administrative Data Research Centre for England (ADRC-E), Farr Institute, University College London, London, UK
[4]Norwich Medical School, University of East Anglia, Norwich, UK
[5]Faculty of Health, Social Care and Education, Anglia Ruskin University, Cambridge, UK

**Correspondence to**
Dr Lion Shahab;
lion.shahab@ucl.ac.uk

## ABSTRACT

**Objectives** To assess the association between tobacco and hazardous alcohol use and sexual orientation and whether such an association could be explained by other sociodemographic characteristics.

**Design** Cross-sectional household survey conducted in 2014–2016.

**Setting** England, UK.

**Participants** Representative English population sample (pooled n=43 866).

**Main outcomes** Sexual orientation identity (lesbian/gay, bisexual, heterosexual, prefer-not-to-say); current tobacco and hazardous alcohol use (defined as Alcohol Use Disorders Identification Test Score ≥8). All outcomes were self-reported.

**Results** Due to interactions between sexual orientation and gender for substance use, analyses were stratified by gender. Tobacco use prevalence was significantly higher among lesbian/gay (women: 24.9%, 95% CI 19.2% to 32.6%; men: 25.9%, 95% CI 21.3% to 31.0%) and bisexual participants (women: 32.4%, 95% CI 25.9% to 39.6%; men: 30.7%, 95% CI 23.7% to 30.7%) and significantly lower for prefer-not-to-say participants in women (15.5%, 95% CI 13.5% to 17.8%) but not men (22.7%, 95% CI 20.3% to 25.3%) compared with heterosexual participants (women: 17.5%, 95% CI 17.0% to 18.0%; men: 20.4%, 95% CI 19.9% to 21.0%; p<0.001 for omnibus test). Similarly, hazardous alcohol use was significantly more prevalent for lesbian/gay (women: 19.0%, 95% CI 14.0% to 25.3%; men: 30.0%, 25.2%–35.3%) and bisexual participants (women: 24.4%, 95% CI 18.7% to 31.3%; men: 24.3%, 95% CI 17.9% to 32.1%) and lower for prefer-not-to-say participants (women: 4.1%, 95% CI 3.0% to 5.4%; men: 13.7%; 95% CI 11.8% to 16.0%) compared with heterosexuals (women: 8.3%, 95% CI 7.9% to 8.7%; men: 18.4%, 95% CI 17.9% to 18.9%; p<0.001 for omnibus test). However, after adjusting for sociodemographic confounders, tobacco use was similar across all sexual orientation groups among both women and men. By contrast, sexual orientation differences in hazardous alcohol use remained even after adjustment among women but not for bisexual and gay men.

**Conclusions** In England, higher rates of tobacco use among sexual minority men and women appear to be attributable to other sociodemographic factors. Higher rates of hazardous alcohol use among sexual minority men

### Strengths and limitations of this study

► Most research assessing sexual orientation and substance use involves small convenience samples, and data from representative samples is scarce. This study is one of only very few studies to investigate this issue in a large representative population survey.
► This study used validated measures of sexual orientation and health behaviours (tobacco and hazardous alcohol use) and controlled for a wide range of confounders to assess the association between them.
► As is the case for all cross-sectional analysis we cannot infer causality from this study and the measure of sexual orientation may not have captured all dimensions of this construct.

may also be attributable to these factors, whereas this is not the case for sexual minority women.

## INTRODUCTION

Knowledge about the health behaviours of sexual minority (lesbian, gay and bisexual (LGB)) groups is necessary for monitoring health inequalities, developing public health policies, allocating resources and targeting high-risk groups for interventions.[1 2]

There is a validated measure of sexual orientation for LGB groups developed by the Office of National Statistics,[3] but despite recent equality and diversity legislation and the inclusion of sexual orientation in the National Health Service (NHS) Equality Delivery System, sexual orientation identity is not regularly monitored and/or reported by NHS organisations or collected in epidemiological research. This may in part be due to the complex nature of sexual orientation, which can be defined and assessed along dimensions of behaviour, identity or attraction or a combination of these.[4] The different dimensions of identity, behaviour and attraction do

not map easily onto each, and even within these dimensions, not all possible manifestations have been explored.[5] Other methodological issues include the fact that, historically, sexual minorities were relatively more hidden due to stigmatisation, resulting in, small, non-representative samples being studied.[6] This also means that a substantial proportion of participants who do not identify as heterosexual many not endorse any other sexual orientation in surveys for various reasons, including the sensitive nature of the topic or conservative attitudes.[7]

Due to this lack of data, the evidence base on health inequalities experienced by sexual minorities is sparse and has only recently begun to be explored. In the UK, LGB sexual identity and behaviour has been assessed in the National Attitudes of Sexual Attitudes and Lifestyles (1990–1991, 1999–2001, 2010–2012), National Statistics Opinions Survey (in 2008–2009), Office for National Statistics (ONS) Integrated Household Survey (from 2009), Longitudinal Study of Young People in England (from 2009), British Cohort Study (2012), Health Survey for England (from 2010), Scottish Health Survey (from 2008), English Longitudinal Study of Ageing (2012/13) and Understanding Society (from 2011/12). In addition, several major cohort studies in the USA have included questions on sexual orientation identity in recent years.[8 9] Yet, due to the methodological difficulties discussed above, most evidence on health behaviours so far has come from small convenience samples, suggesting that sexual minorities are more likely to smoke tobacco or engage in harmful alcohol use.[9] Representative population data across the entire adult age range, especially from non-US samples, are scarce.[5 10 11]

Several possible mechanisms have been proposed that connect sexual orientation identity to health behaviours such as smoking and alcohol use. The concept of minority stress is often invoked to explain how heterosexism and homophobia are internalised,[12] perhaps leading people to self-medicate for psychological distress with cigarettes or other substances such as alcohol. A recent review found support for this theory, particularly in explaining associations with victimisation and substance use.[13] Another factor could be lower levels of self-esteem, well-being and greater propensity to mental health problems in this population brought on by biases towards sexual minorities[5 6 14] which in themselves have been linked to tobacco and alcohol abuse.[15 16] Alternatively, LGB young people, in particular, may be more likely to access adult venues and clubs associated with use of intoxicating substances, including alcohol and illegal drugs.[17] Due to a lack of social support at school, from peers and family, LGB young people may seek this support and socialise in recreational spaces where peer norms encourage engagement in risky health behaviours,[18] or these behaviours may be adopted in order to appear older to gain access to a venue or to fit in.[19] Substance use may be sexually arousing for some individuals and subcultures, particularly when seen to signify masculinity.[20] Other commentators have noted the role of industry in targeting this community.[21]

There is a clear need for population level studies to investigate the extent of differences in tobacco and alcohol use by sexual minorities. For instance, it is possible that apparent differences in prevalence may be an artefact of other sociodemographic differences associated with sexual minorities or the use of non-representative samples. Better describing of tobacco and alcohol use behaviour and identifying effective drivers for change will help reduce health inequalities for the LGB community. This study aimed to assess the prevalence and association of tobacco and hazardous alcohol use with sexual orientation identity. Specifically, the study addressed the following questions:

1. What is the prevalence of tobacco and hazardous alcohol use and distribution of associated use characteristics in a large LGB population sample? Do they differ by gender or from the heterosexual population?
2. Are differences in tobacco and hazardous alcohol use prevalence, if any, between the LGB and heterosexual population sample attenuated when controlling for other sociodemographic covariates?

## METHODS

### Study design and participants

Data were collected using cross-sectional household surveys of representative samples of the population of adults in England. The surveys are part of the ongoing series of Smoking and Alcohol Toolkit Studies, designed to provide tracking information about smoking, alcohol consumption and related behaviours in England. Each month, a new sample of approximately 1700 adults aged ≥16 years complete a face-to-face computer-assisted survey with a trained interviewer (see refs[22 23] for full details). Current data were collected from March 2014 until May 2016 where complete data on tobacco, alcohol use, sexual orientation and other baseline sociodemographic variables were included. A total of 45 423 adults were surveyed in this time period, of whom 43 886 had complete data. The study received ethical approval from the University College London Research Ethics Committee and was carried out in accordance with the ethical principles on human research, as set out in the Declaration of Helsinki.

### Measures

#### Sexual identity

Sexual orientation was assessed by asking respondents to self-identify as (1) bisexual; (2) gay man/homosexual; (3) gay woman/lesbian; (4) heterosexual/straight; (5) prefer-not-to-say. This measure deviates slightly from the recommended ONS item (which also provides an 'Other' option)[3] but produces similar prevalence rates compared with the latest ONS Integrated Household Survey[24] (see table 1).

#### Tobacco use

To assess tobacco use, participants were asked if they (1) smoked cigarettes (including hand-rolled) every day; (2)

**Table 1** Prevalence of self-perceived sexual orientation identity in the Smoking and Alcohol Toolkit Study 2014–2016 and ONS Integrated Household Survey, 2014

| | Toolkit study | ONS |
|---|---|---|
| | % (95% CI) | |
| Heterosexual/straight | 93.3 (93.0 to 93.5) | 92.8 (92.6 to 93.0) |
| Bisexual | 0.7 (0.6 to 0.8) | 0.5 (0.4 to 0.6) |
| Lesbian/gay | 1.1 (1.0 to 1.2) | 1.1 (1.0 to 1.2) |
| PNTS | 4.9 (4.7 to 5.1) | 5.3 (5.2 to 5.4) |
| Other | – | 0.3 (0.3 to 0.3) |

ONS, Office for National Statistics; PNTS, prefer-not-to-say.

smoked cigarettes (including hand-rolled) but not every day; (3) did not smoke cigarettes at all but did smoke tobacco of some kind (eg, pipe or cigar); (4) had stopped smoking completely in the last year; (5) had stopped smoking completely more than a year ago or (6) had never been a smoker (ie, smoked for a year or more). Current smokers were classified as answering 'yes' to (1) to (3) and ex-smokers, classified as answering 'yes' to (4) or (5). Those answering 'yes' to (6) were classified as never-smokers and together with ex-smokers also classified as non-smokers to indicate current tobacco use (yes/no). Additionally, current smokers were asked questions to determine nicotine dependence (measured by heaviness of smoking index (HSI[25]) and strength of urges to smoke (SUTS[26]) as well as motivation to quit (measured by the motivation to stop scale (MTSS[27]). Smokers also provided an estimate of money spent per week on tobacco products, whether they had made at least one serious quit attempt in the last year and whether they had been asked about their smoking, received advice and/or support to stop from their general practitioner (GP).

### Alcohol use

Alcohol use was assessed with the well-established and reliable Alcohol Use Disorders Identification Test questionnaire, with a cut-off ≥8 to indicate hazardous alcohol use (yes/no).[28] Participants who scored as hazardous drinkers were further asked about their urges to drink (adapted from the SUTS) and their motivation to cut down alcohol consumption (adapted from the MTSS). They were also asked how much money they spent per week on alcohol for their own consumption, whether they had made a serious attempt to cut down in the last year and whether they had been asked about their drinking, received advice and/or support to cut down from their GP.

### Covariates

Standard sociodemographic characteristics assessed included age (in years), sex (male/female), ethnicity (white/non-white), marital status (married, civil partnership or living with partner: yes/no), socioeconomic status (SES): ABC1/C2DE; disability: yes/no; education: post-16 (post-high school) qualification: yes/no; England

region: (North/Central/South) and internet use: >daily/≤daily/never.

### Analysis

Data were analysed with IBM SPSS Statistics V.24.0. All data were weighted using the rim (marginal) weighting technique to match an English population profile on the dimensions of age, social grade, region, tenure, ethnicity and working status within sex derived from the English 2011 census, ONS 2013 midyear estimates and a random probability survey conducted in 2014 for the National Readership Survey.[29] Further details on the weighting procedure are reported elsewhere.[30] Simple associations between study groups and continuous and categorical sociodemographic, tobacco use and alcohol use characteristics were assessed with one-way analyses of variance and $\chi^2$ analysis, respectively. General linear models (GLM) with a log-binomial link were used to analyse the association of sexual orientation with the binary outcome variables (current tobacco use; hazardous alcohol consumption) and to calculate prevalence ratios. We included interaction terms for sexual orientation by gender in the GLM to determine if there were systematic differences. Additional stepwise forward and backward logistic regression models were run to determine the individual impact of covariates on the outcome variables, being ordered by the magnitude of changes in variance in the model when covariates are added or removed. Family-wise error rate was corrected using the false discovery rate[31] and multiple comparisons were controlled for using the Sidak correction in post hoc analysis. Only complete cases were analysed and missing data excluded.

### RESULTS

Table 1 provides information about the prevalence of sexual orientation in this sample and table 2 provides a breakdown of associated sociodemographic characteristics. Bisexual participants were significantly younger than other participants, followed by lesbian/gay participants, with no differences in age between heterosexual and prefer-not-to-say participants. Participants who self-identified as lesbian/gay were less likely to be female and more likely to be white, and prefer-not-to-say participants were less likely to be white than other participants. Heterosexual participants were least likely and bisexual participants were most likely to be single. Bisexual and prefer-not-to-say participants were more likely to be from lower social grades, and lesbian/gay participants were least likely to be from lower social grades. Lesbian/gay participants were also most likely to have post-16 qualification and bisexuals were more likely than all other groups to report a disability. There were some small regional differences, with lesbian/gay participants being most likely to reside in South England and prefer-not-to-say participants most likely to live in Central England. All groups differed in terms of internet access, with more than daily use being most common among lesbian/gay participants, followed

**Table 2** Sociodemographic characteristics of sample by sexual orientation identity

| Characteristic | Total (n=44 030) | Heterosexual (n=41 058) | Bisexual (n=316) | Lesbian/Gay (n=498) | Prefer-not-to-say (n=2158) | p Value |
|---|---|---|---|---|---|---|
| Mean (SD) age (years) | 47.1 (18.6) | 47.3 (18.6)[a] | 32.5 (14.6)[b] | 39.0 (15.4)[c] | 47.6 (19.1)[a] | <0.001 |
| % (N) women | 50.9 (22 406) | 51.1 (20 962)[a] | 55.7 (176)[a] | 37.1 (185)[b] | 50.2 (1083)[a] | <0.001 |
| % (N) white | 86.7 (38 192) | 87.0 (35 713)[a] | 87.4 (277)[a] | 96.0 (478)[b] | 79.9 (1724)[c] | <0.001 |
| % (N) single | 41.4 (18 244) | 40.4 (16 600)[a] | 64.4 (204)[b] | 55.6 (277)[b, c] | 53.9 (1163)[c] | <0.001 |
| % (N) social grade C2DE | 45.4 (19 971) | 45.1 (18 533)[a] | 52.7 (167)[b] | 33.4 (166)[c] | 51.2 (1105)[b] | <0.001 |
| % (N) No post-16 qualification | 34.3 (15 082) | 34.5 (14 156)[a] | 28.4 (90)[a] | 19.5 (97)[b] | 34.2 (739)[a] | <0.001 |
| % (N) with disability | 10.3 (4541) | 10.2 (4188)[a] | 19.9 (63)[b] | 13.5 (67)[a] | 10.3 (223)[a] | <0.001 |
| % (N) region | | [a] | [a] | [b] | [c] | <0.001 |
| North | 28.8 (12 665) | 29.3 (12 043) | 33.4 (106) | 32.5 (162) | 16.4 (354) | |
| Central | 30.0 (13 209) | 29.8 (12 243) | 31.5 (100) | 18.9 (94) | 35.8 (772) | |
| South | 41.2 (18 158) | 40.9 (16 773) | 35.0 (110) | 48.6 (242) | 47.8 (1032) | |
| % (N) internet access | | [a] | [b] | [c] | [d] | <0.001 |
| Never | 12.8 (5651) | 12.7 (5231) | 9.5 (30) | 3.4 (17) | 17.3 (373) | |
| ≤Daily | 19.8 (8718) | 20.0 (8218) | 12.0 (38) | 11.0 (55) | 18.9 (407) | |
| >Daily | 67.4 (29 661) | 67.2 (27 609) | 78.5 (248) | 85.5 (426) | 63.9 (1378) | |

[a, b, c, d]Different letters for groups in each row indicate significant differences between these groups after controlling for multiple comparisons (p<0.05), same letters indicate no group differences; please note that weighted data are shown.

by bisexual and heterosexual participants and being least common among prefer-not-to-say participants.

### What is the prevalence of tobacco and hazardous alcohol use and distribution of associated use characteristics in a large LGB population sample? Do they differ by gender or from the heterosexual population?

As there was a significant interaction between sexual orientation and gender for hazardous alcohol use (Wald $X^2(3)$=21.46, p<0.001) and a near significant interaction for tobacco use (Wald $X^2(3)$=7.76, p=0.051), further analysis on health behaviours was stratified by gender (see tables 3 and 4).

Differences in tobacco and hazardous alcohol use as a function of sexual orientation identities appeared more pronounced among women than men (figure 1A,B) as indicated by a better fit of the simple model for women than men regarding both tobacco (Akaike Information Criterion (AIC) 35.7 vs 36.4) and hazardous alcohol use (AIC 33.4 vs 35.9).

Irrespective of gender, tobacco use was most prevalent among bisexual participants who were nearly twice as likely to smoke as their heterosexual counterparts (for men: 30.7%, 95% CI 23.7% to 30.7% versus 20.4%, 95% CI 19.9% to 21.0%; prevalence ratio (PR) 1.61, 95% CI 1.28 to 2.03; for women: 32.4%, 95% CI 25.9% to 39.6% versus 17.5%, 95% CI 17.0% to 18.0%; PR 1.87, 95% CI 1.52 to 2.29). Similarly, those who self-identified as lesbian/gay were also more likely to smoke than those with a heterosexual identity (for men: 25.9%, 95% CI 21.3% to 31.0%; PR 1.30, 95% CI 1.08 to 1.56; for women: 24.9%, 95% CI 19.2% to 32.6%; PR 1.41, 95% CI 1.11 to 1.81). However,

while for male prefer-not-to-say participants, tobacco use prevalence was similar to heterosexual participants (22.7%, 95% CI 20.3% to 25.3%; PR 1.07, 95% CI 0.96 to 1.20), it was significantly lower for female prefer-not-to-say participants (15.5%, 95% CI 13.5% to 17.8%; PR 0.87, 95% CI 0.76 to 1.00; figure 1A).

The pattern for hazardous alcohol across sexual orientation identities also differed between men and women (figure 1B). Irrespective of gender, prefer-not-to-say participants had the lowest prevalence of hazardous alcohol use, at nearly half the prevalence observed among heterosexual participants (for men: 13.7%; 95% CI 11.8% to 16.0% vs 18.4%, 95% CI 17.9% to 18.9%; PR 0.74, 95% CI 0.63 to 0.86; for women: 4.1%, 95% CI 3.0% to 5.4% vs 8.3%, 95% CI 7.9% to 8.7%; PR 0.48, 95% CI 0.35 to 0.64). However, while for men, hazardous alcohol use was greatest among those who self-identified as gay, at nearly twice the prevalence of heterosexual men (30.0%, 95% CI 25.2% to 35.3%; PR 1.61, 95% CI 1.35 to 1.91), followed by bisexual men (24.3%, 95% CI 17.9% to 32.1%; PR 1.41, 95% CI 1.07 to 1.85), a different pattern was observed in women. Bisexual women had the highest prevalence rates, being more than three times as likely as heterosexual women to engage in hazardous alcohol use (24.4%, 95% CI 18.7% to 31.3%; PR 3.08, 95% CI 2.39 to 3.95), followed by lesbian participants (19.0%, 95% CI 14.0% to 25.3%; PR 2.18, 95% CI 1.60 to 2.96).

Table 3A,B provide information about tobacco use and hazardous alcohol use characteristics by sexual orientation identity in women and men. Among current or recent tobacco users, bisexual women appeared less dependent

**Table 3** Tobacco and hazardous alcohol use characteristics by sexual orientation identity in (A) women and (B) men

| | Total | Heterosexual | Bisexual | Lesbian/Gay | Prefer-not-to-say | p Value |
|---|---|---|---|---|---|---|
| **A. Women** | | | | | | |
| Tobacco users* | (n=4243) | (n=3952) | (n=60) | (n=50) | (n=181) | |
| Mean (SD) cigarettes per day† | 10.8 (7.7) | 10.8 (7.6) | 8.1 (7.8) | 10.8 (10.3) | 10.8 (8.5) | 0.091 |
| % (N) primarily RYO use† | 43.1 (1532) | 42.4 (1406)[a] | 65.3 (32)[b] | 50.0 (20)[a, b] | 48.1 (74)[a, b] | 0.005 |
| Mean (SD) HSI | 1.8 (1.5) | 1.8 (1.5)[a] | 1.2 (1.4)[b] | 1.6 (1.9)[a, b] | 1.7 (1.4)[a, b] | 0.011 |
| Mean (SD) urge to smoke | 2.9 (1.2) | 2.9 (1.2) | 2.6 (1.1) | 2.8 (1.3) | 2.7 (1.1) | 0.036 |
| Mean (SD) MTSS‡ | 3.3 (2.0) | 3.3 (2.0) | 3.3 (2.0) | 3.6 (2.0) | 3.2 (2.0) | 0.775 |
| Mean (SD) spent per week (£)‡ | 22.5 (19.4) | 22.7 (19.4)[a] | 14.4 (12.9)[b] | 26.5 (27.9)[a] | 20.4 (17.0)[a, b] | 0.003 |
| % (N) quit attempt last year | 36.0 (1489) | 36.3 (1397) | 31.7 (19) | 40.0 (20) | 29.9 (53) | 0.281 |
| % (N) GP advice | 39.9 (1691) | 40.2 (1587) | 25.4 (15) | 32.0 (16) | 40.3 (73) | 0.087 |
| Hazardous alcohol users§ | (n=4351) | (n=4100) | (n=59) | (n=74) | (n=118) | |
| Mean (SD) urge to drink | 1.5 (0.9) | 1.5 (0.9) | 1.6 (1.2) | 1.8 (1.2) | 1.7 (1.0) | 0.011 |
| Mean (SD) Motivation to cut down drinking | 1.9 (1.6) | 1.8 (1.6) | 1.9 (1.4) | 2.1 (1.9) | 2.1 (1.9) | 0.184 |
| Mean (SD) spent per week (£) | 14.7 (14.1) | 14.6 (14.0) | 16.5 (16.6) | 19.0 (16.5) | 14.8 (11.4) | 0.050 |
| % (N) Attempt to cut-down last year | 16.3 (676) | 16.1 (629) | 16.1 (9) | 20.3 (15) | 21.7 (23) | 0.355 |
| % (N) GP advice | 7.3 (318) | 7.3 (301) | 10.2 (6) | 6.8 (5) | 5.0 (6) | 0.647 |
| **B. Men** | | | | | | |
| Tobacco users* | (n=4843) | (n=4426) | (n=51) | (n=87) | (n=269) | |
| Mean (SD) cigarettes per day† | 11.8 (8.7) | 11.8 (8.6) | 14.3 (17.9) | 11.5 (10.1) | 11.5 (8.1) | 0.256 |
| % (N) primarily RYO use† | 51.0 (2034) | 50.7 (1858) | 60.0 (21) | 42.7 (32) | 58.0 (123) | 0.057 |
| Mean (SD) HSI | 1.8 (1.5) | 1.8 (1.5) | 2.1 (1.8) | 1.9 (1.7) | 1.9 (1.5) | 0.670 |
| Mean (SD) urge to smoke | 2.8 (1.2) | 2.8 (1.2) | 2.6 (1.2) | 3.0 (1.1) | 2.9 (1.2) | 0.103 |
| Mean (SD) MTSS‡ | 3.1 (1.9) | 3.1 (1.9) | 3.2 (2.2) | 3.1 (2.0) | 2.9 (1.9) | 0.344 |
| Mean (SD) Spent per week (£)‡ | 23.0 (21.4) | 23.2 (21.6) | 20.3 (18.4) | 21.9 (17.7) | 21.0 (20.0) | 0.350 |
| % (N) quit attempt last year | 32.1 (1508) | 32.1 (1380) | 41.2 (21) | 39.5 (34) | 28.2 (73) | 0.115 |
| % (N) GP advice | 33.1 (1600) | 32.8 (1450) | 48.1 (25) | 32.2 (28) | 36.1 (97) | 0.086 |
| Hazardous alcohol users§ | (n=7768) | (n=7285) | (n=58) | (n=154) | (n=272) | |
| Mean (SD) urge to drink | 1.7 (1.0) | 1.7 (1.0)[a] | 2.0 (1.3)[a, b] | 1.7 (1.2)[a, b] | 2.0 (1.0)[b] | <0.001 |
| Mean (SD) Motivation to cut down drinking | 1.7 (1.4) | 1.7 (1.4)[a] | 1.9 (1.7)[a, b] | 2.1 (1.7)[b] | 1.7 (1.4)[a, b] | 0.005 |
| Mean (SD) spent per week (£) | 21.8 (20.2) | 21.6 (20.2) | 21.3 (21.5) | 25.4 (23.0) | 24.2 (17.1) | 0.027 |
| % (N) attempt to cut-down last year | 14.5 (1086) | 14.5 (1009)[a] | 10.5 (6)[a, b] | 23.3 (35)[b] | 14.5 (36)[a, b] | 0.019 |
| % (N) GP advice | 9.9 (766) | 10.1 (734) | 8.6 (5) | 8.4 (13) | 5.2 (14) | 0.056 |

[a, b, c, d]Different letters for groups in each row indicate significant differences between these groups after controlling for multiple comparisons (p<0.05), same letters indicate no group differences; please note that weighted data are shown.
*Current or past year tobacco users.
†Current or past year cigarette smokers only.
‡Current tobacco users only.
§AUDIT score ≥8 or AUDIT-C ≥5 score.
GP, general practitioner; HSI, heaviness of smoking index; MTSS, Motivation to Stop Scale; RYO, roll-your-own.

than heterosexual women (as measured by the heaviness of smoking index) were more likely to smoke roll-your-own cigarettes and consequently spent less money on tobacco than other groups (table 3A). There were no group differences in hazardous alcohol use characteristics among women.

By contrast, among men, there were notable differences between sexual orientation identities in hazardous alcohol use but not tobacco use characteristics (table 3B). Hazardous drinkers in the prefer-not-to-say group reported greater urges to drink, and gay men reported greater motivation to cut down on alcohol and had made

more attempts to cut down in the last year compared with heterosexual men.

## Are differences in tobacco and hazardous alcohol use prevalence, if any, between the LGB and heterosexual population attenuated when controlling for other covariates?

The associations of sexual orientation identity with tobacco and hazardous alcohol use were re-examined in women and men, controlling for sociodemographic variables (table 4A,B). Younger age, being white, single, from a lower SES group, lacking post-16 qualifications, reporting a disability and living in Northern England

**Table 4** Factors associated with tobacco and hazardous alcohol use in (A) women and (B) men

| Factor | Tobacco use | | | | Hazardous alcohol use | | | |
|---|---|---|---|---|---|---|---|---|
| | PR (95% CI) | p Value | aPR (95% CI) | p Value | PR (95% CI) | p Value | aPR (95% CI) | p Value |
| **A. Women** | | | | | | | | |
| Age | 0.99 (0.98 to 0.99) | <0.001 | 0.98 (0.98 to 0.98) | <0.001 | 0.97 (0.97 to 0.97) | <0.001 | 0.98 (0.97 to 0.98) | <0.001 |
| White | 2.97 (2.59 to 3.42) | <0.001 | 3.57 (3.10 to 4.11) | <0.001 | 4.14 (3.24 to 5.29) | <0.001 | 4.90 (3.82 to 6.28) | <0.001 |
| Single | 1.51 (1.43 to 1.59) | <0.001 | 1.27 (1.20 to 1.34) | <0.001 | 1.49 (1.37 to 1.63) | <0.001 | 1.47 (1.35 to 1.59) | <0.001 |
| Social grade C2DE | 2.01 (1.94 to 2.17) | <0.001 | 1.69 (1.59 to 1.80) | <0.001 | 0.81 (0.74 to 0.88) | <0.001 | 0.85 (0.78 to 0.93) | <0.001 |
| No post-16 qualification | 1.57 (1.49 to 1.66) | <0.001 | 1.39 (1.31 to 1.47) | <0.001 | 0.59 (0.54 to 0.65) | <0.001 | 0.80 (0.73 to 0.89) | <0.001 |
| With disability | 1.39 (1.29 to 1.50) | <0.001 | 1.33 (1.24 to 1.43) | <0.001 | 0.84 (0.73 to 0.97) | 0.019 | 1.13 (0.98 to 1.30) | 0.092 |
| Region | | <0.001 | | <0.001 | | <0.001 | | <0.001 |
| North (ref) | 1 | | 1 | | 1 | | 1 | |
| Central | 0.74 (0.69 to 0.79) | | 0.83 (0.78 to 0.89) | | 0.37 (0.33 to 0.41) | | 0.45 (0.40 to 0.51) | |
| South | 0.67 (0.63 to 0.71) | | 0.86 (0.81 to 0.91) | | 0.47 (0.42 to 0.51) | | 0.59 (0.53 to 0.64) | |
| Internet access | | 0.125 | | 0.063 | | <0.001 | | <0.001 |
| Never (ref) | 1 | | 1 | | 1 | | 1 | |
| ≤Daily | 0.95 (0.86 to 1.04) | | 0.94 (0.86 to 1.04) | | 2.08 (1.62 to 2.66) | | 1.47 (1.14 to 1.89) | |
| >Daily | 1.02 (0.94 to 1.10) | | 0.89 (0.81 to 0.99) | | 4.73 (3.80 to 5.89) | | 2.26 (1.78 to 2.88) | |
| Sexual orientation identity | | <0.001 | | 0.466 | | <0.001 | | <0.001 |
| Heterosexual (Ref) | 1 | | 1 | | 1 | | 1 | |
| Bisexual | 1.87 (1.52 to 2.29) | | 1.04 (0.88 to 1.24) | | 3.08 (2.39 to 3.95) | | 1.63 (1.30 to 2.04) | |
| Lesbian | 1.41 (1.11 to 1.81) | | 1.14 (0.93 to 1.39) | | 2.18 (1.60 to 2.96) | | 1.37 (1.02 to 1.82) | |
| Prefer-not-to-say | 0.87 (0.76 to 1.00) | | 0.94 (0.83 to 1.08) | | 0.48 (0.35 to 0.64) | | 0.61 (0.45 to 0.81) | |
| **B. Men** | | | | | | | | |
| Age | 0.98 (0.98 to 0.99) | <0.001 | 0.98 (0.98 to 0.98) | <0.001 | 0.99 (0.98 to 0.99) | <0.001 | 0.99 (0.98 to 0.99) | <0.001 |
| White | 1.19 (1.10 to 1.28) | <0.001 | 1.44 (1.34 to 1.55) | <0.001 | 5.53 (4.68 to 6.54) | <0.001 | 6.04 (5.10 to 7.15) | <0.001 |
| Single | 1.62 (1.54 to 1.70) | <0.001 | 1.16 (1.10 to 1.23) | <0.001 | 1.37 (1.30 to 1.45) | <0.001 | 1.21 (1.14 to 1.28) | <0.001 |
| Social grade C2DE | 1.93 (1.83 to 2.04) | <0.001 | 1.55 (1.47 to 1.65) | <0.001 | 0.81 (0.77 to 0.86) | <0.001 | 0.87 (0.82 to 0.92) | <0.001 |
| No post-16 qualification | 1.62 (1.54 to 1.70) | <0.001 | 1.38 (1.31 to 1.46) | <0.001 | 0.77 (0.72 to 0.82) | <0.001 | 0.88 (0.82 to 0.94) | <0.001 |
| With disability | 1.39 (1.29 to 1.50) | <0.001 | 1.35 (1.26 to 1.45) | <0.001 | 0.91 (0.82 to 1.00) | 0.059 | 1.01 (0.92 to 1.12) | 0.797 |
| Region | | <0.001 | | 0.001 | | <0.001 | | <0.001 |
| North (ref) | 1 | | 1 | | 1 | | 1 | |
| Central | 0.85 (0.76 to 0.91) | | 0.89 (0.84 to 0.95) | | 0.46 (0.43 to 0.50) | | 0.54 (0.50 to 0.58) | |

Continued

**Table 4** Continued

| Factor | Tobacco use | | | | Hazardous alcohol use | | | |
|---|---|---|---|---|---|---|---|---|
| | PR (95% CI) | p Value | aPR (95% CI) | p Value | PR (95% CI) | p Value | aPR (95% CI) | p Value |
| South | 0.81 (0.75 to 0.87) | | 0.94 (0.89 to 1.00) | | 0.61 (0.57 to 0.65) | | 0.70 (0.66 to 0.75) | |
| Internet access | | <0.001 | | <0.001 | | <0.001 | | <0.001 |
| Never (ref) | 1 | | 1 | | 1 | | 1 | |
| ≤Daily | 0.83 (0.77 to 0.91) | | 0.83 (0.76 to 0.90) | | 1.34 (1.17 to 1.54) | | 1.09 (0.95 to 1.25) | |
| >Daily | 0.86 (0.80 to 0.92) | | 0.76 (0.70 to 0.82) | | 1.94 (1.73 to 2.19) | | 1.31 (1.15 to 1.50) | |
| Sexual orientation identity | | <0.001 | | 0.517 | | <0.001 | | 0.031 |
| Heterosexual (Ref) | 1 | | 1 | | 1 | | 1 | |
| Bisexual | 1.61 (1.28 to 2.03) | | 1.15 (0.93 to 1.42) | | 1.41 (1.07 to 1.85) | | 1.07 (0.83 to 1.38) | |
| Gay | 1.30 (1.08 to 1.56) | | 1.06 (0.89 to 1.26) | | 1.61 (1.35 to 1.91) | | 1.10 (0.94 to 1.30) | |
| Prefer-not-to-say | 1.07 (0.96 to 1.20) | | 1.03 (0.92 to 1.14) | | 0.74 (0.63 to 0.86) | | 0.82 (0.71 to 0.95) | |

aPR, adjusted prevalence (risk) ratio; PR, prevalence (risk) ratio.

were all independently associated with current tobacco use as was, in men only, lack of internet access. As determined by stepwise forward and backward logistic regression, the association between sexual orientation with smoking status was removed in both women and men after including age, ethnicity, marital status (for women only), SES status and educational attainment into the model; variables are listed in order of impact on changes in variance explained.

By contrast, the association of sexual orientation identity with hazardous alcohol use persisted in women but less so among men, even after controlling for sociodemographic characteristics (table 4). Compared with heterosexual women, women who self-identified as bisexual (PR 1.63; 95% CI 1.30 to 2.04) or lesbian (PR 1.37; 95% CI 1.02 to 1.82) were more likely to engage in hazardous alcohol use in adjusted analysis, while prefer-not-to-say women were less likely to do so (PR 0.61; 95% CI 0.45 to 0.81; table 4A). By contrast, among men, differences only persisted for prefer-not-to-say men who remained less likely to engage in hazardous drinking (PR 0.82; 95% CI 0.71 to 0.95) than heterosexuals. However, bisexual (PR 1.07, 95% CI 0.83 to 1.38) and gay men (OR 1.10, 95% CI 0.94 to 1.30) no longer differed from heterosexual men in hazardous drinking after adjusting for sociodemographic covariates (table 4B). Being younger, white, single, from a higher SES group and having post-16 qualifications as well as accessing the internet and being from Northern England were all independently associated with hazardous drinking in both men and women.

## DISCUSSION

Our study found that in England, lesbian, gay and bisexual men and women appear to have higher rates of tobacco and hazardous alcohol use. This is in agreement with previous studies of young people and adults, reporting greater risk of sexual minorities engaging in tobacco and hazardous alcohol use than their heterosexual counterparts.[9 32–34] Those who elected not to disclose their sexual orientation were less likely to engage in these risky health behaviours. In general, it appears that differences were more pronounced among women than men, as has been previously reported.[35] However, contrary to previous work, we found that differences between sexual orientation identities in tobacco use disappeared when sociodemographic variables were taken into account. Disparities in hazardous alcohol use across sexual orientation identities were also somewhat attenuated when controlling for these covariates, but mainly among men, and remained significant in women.

Differences in tobacco use between LGB and heterosexual participants in this study appear to be explained in the most part by underlying variations in major sociodemographic characteristics. However, it is important to remember that tobacco use disparities remain in this

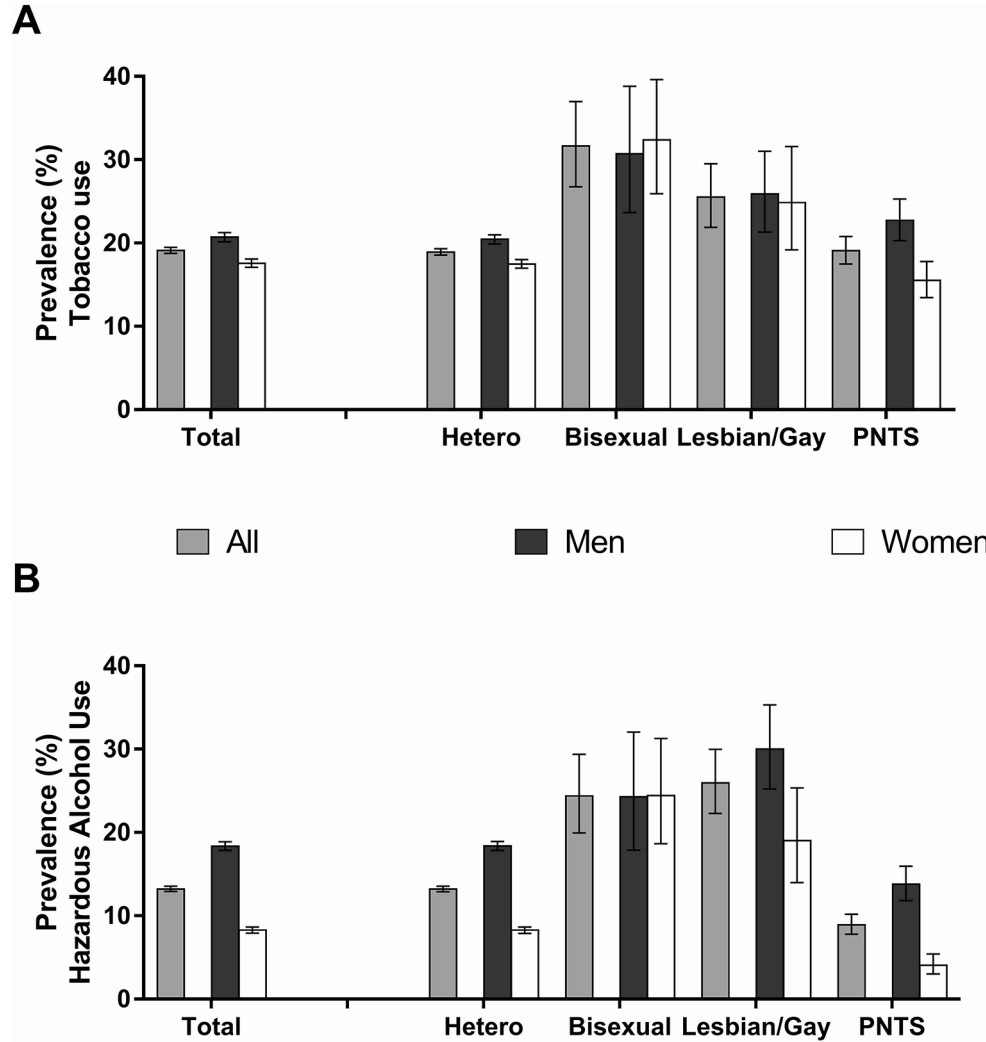

**Figure 1** Prevalance of (A) tobacco use (B) hazarddous alcohol use by gender and sexual orientation identity. PNTS, prefer-not-to-say; Error bars are 95% CIs; All data are unadjusted.

group,[10 32 36] even if these may not be attributable to factors specifically associated with sexual orientation, and further work needs to be done to support minority groups to reduce tobacco consumption. By contrast, the association of hazardous alcohol use with being a sexual minority largely persisted after controlling for major sociodemographic confounders, but only among women. This persistence suggests that there may exist specific influences that put lesbian and bisexual women at greater risk as has been previously reported for both alcohol use[11 37] and other health behaviours.[38] While our findings replicate the existing evidence of a higher risk profile in sexual minority groups,[14 34 39] they underline the need for gender-specific tailoring of health messages to account for the differences within the LGB community. This is for instance shown by the finding that gay but not lesbian hazardous drinkers reported both greater motivation to cut down on alcohol and had more quit attempts compared with heterosexual hazardous drinkers.

The finding that the association of sexual orientation identity with tobacco use was attenuated after controlling for sociodemographic characteristics, in particular age,

ethnicity, marital status (among women), socioeconomic group and educational attainment, could be due to several reasons. Self-identification as LGB is greater in younger than older age groups[24] as are smoking rates. Participants who identified as lesbian/gay were more likely to be white and LGB participants were more likely to be single, both factors associated with greater smoking rates in the general population.[24 40] However, it is more difficult to explain the role of deprivation (as measured by education or socioeconomic group). Generally, deprivation is strongly and positively associated with tobacco use[41] and the association of greater deprivation with tobacco use was as expected for bisexual participants, particularly bisexual women, who were more likely to smoke, use cheaper roll-your-own cigarettes and spend less on tobacco. Moreover, this group was also younger than other groups, including lesbian/gay participants, which would explain both the association with lower dependence (older smokers are more dependent) and deprivation (age inversely relates to income). By contrast, lesbian/gay participants had both lower levels of deprivation and higher smoking rates than heterosexuals, suggesting that, unlike for other

groupings (eg, by ethnicity or gender[42]), the association of deprivation with tobacco use may be more complicated in sexual orientation minorities. Smoking as a health behaviour may be performed differently in these groups, both compared with each other and in comparison with heterosexuals.[8] This issue deserves further investigation.

More generally, the results may suggest a genuine shift in the environment encountered by sexual minorities. The social-ecological model[43] posits that interactions with the environment determines risk behaviour. Therefore reduced differences in tobacco use could indicate that the environment for sexual minorities and the majority is becoming increasingly similar. Moreover, the introduction of 'smoke-free' laws may have had a disproportionately greater impact on tobacco use among sexual minorities as they are more likely to spend time in social, recreational spaces[18] in which smoking used to be the norm.

The persistent association with hazardous alcohol use among sexual minority participants in this study, specifically lesbian and bisexual women, highlights the negative behavioural consequences experienced by social minorities, not easily explained by general sociodemographic covariates. One potential explanatory factor not explored in the current analysis is the known increased risk of mental health problems in the LGB community[5] which is also associated with problem drinking.[44] Other factors include the use of recreational spaces by sexual minorities where alcohol drinking may be the norm. However, the finding that the association with hazardous alcohol use is particularly strong among sexual minority women suggests that other gender-specific influences may be at work. For instance, it may be an expression of gender non-conformity (ie, to go against stereotypical views of female vs male drinking behaviour[45]) or it may reflect the fact that women are more likely to experience double discrimination[46] which may increase the propensity to engage in risky health behaviours as a coping mechanism.

It is interesting to note that participants who preferred not to disclose their sexual orientation were less likely to engage in hazardous alcohol use, whereas no differences were observed in this group in relation to tobacco use. As there was a larger proportion of non-white participants in this group, this may be explained in part by different cultural and religious norms and stigma imposed on people; that is, many religions that adopt a negative stance towards sexual minorities, which may lead to non-disclosure of sexual minority identity,[47] often also have punitive views on alcohol, resulting in lower consumption.[48] Similarly, while there are few ethnic and cultural disparities in male (though not female) smoking rates, alcohol use is differently patterned by ethnicity with some ethnic minorities drinking less.[49] Against these barriers to drinking, self-selection may therefore explain why those prefer-not-to-say participants who do drink are more dependent.

This study had a number of limitations. The cross-sectional nature of the design makes it difficult to assess causal

pathways. We did attempt to reduce the risk of unmeasured bias by controlling for a range of known sociodemographic confounders for tobacco and hazardous alcohol use. While this sample was representative of the general population of England and results therefore likely generalise to other high-income countries, the subgroup of participants with a minority sexual orientation identity was relatively small and so the study would not have been powered to detect smaller and more subtle differences. As has been previously discussed, sexuality can be measured in different ways. Here, a measure of sexual orientation identity was used, as this is argued to be the most relevant dimension to investigate the relationship of sexual orientation with disadvantage.[50] Future research should consider investigating this question using measures of behaviour and attraction: groups who do not choose to identify as LGB but carry out same-sex behaviour for instance may be quite different. Not everyone is comfortable adopting an identity label and some may have not wanted to select one of the options offered by the ONS question. Lastly, no objective measure of tobacco and alcohol use was taken; however, both behaviours were assessed using validated and reliable scales and low demand, anonymous studies tend to provide relatively unbiased results.[51]

In conclusion, sexual orientation disparities in tobacco and hazardous alcohol use exist in England, with LGB people exhibiting greater levels of risky health behaviours. However, differences in tobacco and hazardous alcohol use appear mainly associated with general sociodemographic differences in men, whereas differences in hazardous alcohol use, but not tobacco use, persist in women after controlling for sociodemographic characteristics. Further research is now needed to consider the explanatory factors and to develop interventions to remove health inequalities in these populations.

**Contributors** CM, JB and RW conceived the original idea for this study. CM obtained funding. JB and RW managed the day-to-day running of the study. LS undertook the data analyses and wrote the initial draft with further input from all authors. LS is guarantor for this article. LS, JB, GHJ, SM, JS, RW and CM read, reviewed and approved the final version. All researchers listed as authors are independent from the funders and all final decisions about the research were taken without constraint by the investigators. LS had full access to all the data in the study and had final responsibility for the decision to submit for publication.

**Funding** We are grateful to Cancer Research UK who funded this analysis (C52095/A18725). The Smoking Toolkit Study is primarily funded by Cancer Research UK (C1417/A14135; C36048/A11654; C44576/A19501; C1417/A22962) and has previously also been funded by Pfizer, GSK and the Department of Health. The National Institute for Health Research (NIHR) School for Public Health Research funded data collection for the Alcohol Toolkit Study (SPHR-SWP-ALC-WP5). JB and RW are funded by Cancer Research UK (C1417/A22962). SPHR is a partnership between the universities of Sheffield; Bristol; Cambridge; Exeter; UCL, The London School for Hygiene and Tropical Medicine; the LiLaC collaboration between the universities of Liverpool and Lancaster and Fuse; The Centre for Translational Research in Public Health, a collaboration between Newcastle, Durham, Northumbria, Sunderland and Teesside universities. The views expressed are those of the authors(s) and not necessarily those of the NHS, NIHR or Department of Health.

**Competing interests** LS has received a research grant and honoraria for a talk and travel expenses from a Pfizer, manufacturer of smoking cessation medications. JB has received unrestricted research funding from Pfizer to study smoking cessation. RW has received travel funds and hospitality from, and undertaken

research and consultancy for, pharmaceutical companies that manufacture or research products aimed at helping smokers to stop. GHJ receives royalties from 'Introduction to Research Methods and Data Analysis in Psychology' (2013) and 'Psychometric Assessment, Statistics and Report Writing' (2013), both published by Pearson and 'Introduction to Research Methods and Data Analysis in the Health Sciences'. He has also worked as a consultant for various universities and Public Health England. His substantive contribution to the manuscript was prior to June 2016. CM, SM and JS have no competing interests.

**Ethics approval** UCL REC.

**Provenance and peer review** Not commissioned; externally peer reviewed.

**Data sharing statement** Headline data from the Toolkit Study are published monthly at http://www.smokinginengland.info/. Data will be made available upon request.

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
