## [Reviewer comments · BMJ Open]

ARTICLE DETAILS

TITLE (PROVISIONAL)	Sexual orientation identity and tobacco and hazardous alcohol use: findings from a cross-sectional English population survey
AUTHORS	Shahab, Lion; Brown, Jamie; Hagger-Johnson, Gareth; Michie, Susan; Semlyen, Joanna; West, Robert; Meads, Catherine

VERSION 1 - REVIEW

REVIEWER	Ryan David Kennedy Johns Hopkins University
REVIEW RETURNED	20-Feb-2017

GENERAL COMMENTS	Overall comments – Important paper – clearly adds to the literature with the robust study design; well written. Findings are important too – given the ability to adjust for important sociodemographic factors. Specific comments – mostly just suggestions for writing - Abstract - good Strengths and limitations – second point – consider re-wording – just seemed off with the listing of goals – maybe something like “...with the aim to inform health policy, support strategic allocation of resources and, if necessary, to develop targeted interventions. Introduction – Is the issue (paragraph starting ‘ there is a validated measure’... - that orientation is not reported – or the health disparities from this community are not sufficiently explored? This is a critical idea – and I would ask the authors to take a bit more space here to spell out the issues clearly before jumping to reasons why this might be the case. Paragraph “several possible mechanisms” – authors propose that LGB young people may be more likely to access adult venues where alcohol and cigarettes are easily available – no reference provided; England has been smoke-free for many years – is this an outdated concept (for cigarettes? I can see the argument for alcohol) -? Provide references if possible. Sentence - or the use non-representative data... (the use of non-representative data?) Paragraph “There is clear need for” – consider “There is a clear need for population level studies to investigate [then state what you want to investigate]. Do you need to explain in the introduction why Trans folks are not included? (LGB population only?) – It is highlighted partially in the limitations (and strengths) section – but suggest being explicit.
--

	Methods – Table 1 – seems odd to have a table in the Methods section – but perhaps more of an editorial decision (also - no CIs for the data from ONS?) Post-16 – this might be specific to the UK? Perhaps explain? Results – You discuss Table 2 in the results before discussing Table 1 – consider how to change this – Discussion – Similar findings have been revealed when dealing with other groups – I encourage the authors to include (briefly) how SES drives smoking rates/prevalence among groups based on race (for example) – Concluding paragraph – authors presented evidence that tobacco and alcohol use were higher in the LGB community – but argued that data were not historically representative – so would caution starting the conclusion with saying these levels of use persist. Adding more detail about the PNTS group – for readers not familiar with this group – perhaps in the introduction it is worth outlining why some folks might not wish to say – given that this group by numbers is larger than the other groups of interest. Further discussing how to best interpret this group in the discussion would be helpful too. Excellent paper – will make a real contribution
--	--

REVIEWER	Gary Chan University of Queensland
REVIEW RETURNED	11-Apr-2017

GENERAL COMMENTS	The authors examined the association between sexual orientation and tobacco/alcohol use in a large nationally representative sample. I have read this paper with interest. The introduction is well-written and the rationale is sound. I think there are rooms for improvement in the analyses and I have some recommendation below. First, the authors briefly reported that there was a significant interaction between gender and sexual orientation, and then stratify most of the analyses by gender. I think this is an important finding and a more proper way to follow up is to include this interaction term in the final logistic regression models. The authors can then calculate the marginal probabilities of use to see how the associations between sexual orientation and tobacco/alcohol use differ by gender. This also allows the authors to test the differences more rigorously. For example, statement like “Differences in tobacco and hazardous alcohol use as a function of sexual orientation identities were more pronounced among women than men” can be tested more rigorously by examining the interaction and follow up with tests based on marginal probabilities (or other tests similar to the simple slope analysis in linear regression). From the odds ratio estimates, it seems that the biggest differences is between male and female is the comparison between bisexual and heterosexual individual (For example, in the unadjusted hazardous drinking model, the odds ratio for bisexual for female and male were 3.75 and 1.64 respectively. But the odds ratio for homosexual for female
---

	and male were very similar – 2.53 and 2.23 respectively and there are large overlapping between the CI of these two estimates). At the moment the authors just described the differences without actually testing if the differences are statistically significant. Another advantage of this approach is that numbers of regression models run can be reduced by half, giving a more parsimonious description of the data. Second, the findings that the association between tobacco use and sexual orientation became non-significant after adjusting for socio-demographic factors is also very interesting. The authors have put forward a few possible explanations in the discussion. I think that the authors can potentially more accurately pinpoint which factors can account for the largest attenuation of effect. This can be done by entering the socio-demographic variables one by one into the logistic regression models. Some statements in the discussion are not accurate. For example, “In general, it appears that differences were more pronounced among women than men, as has been previously reported”. It seems that the only gender differences were in the comparison between bisexual and heterogeneous individuals, but the gender differences in the comparison between homogenous and heterogeneous individuals were similar. This can be tested using the authors’ data and needs to be clarified in the discussion. The statement “Moreover, the importance of targeting health messages to lesbian/bisexual women differently to gay/bisexual men is underlined by the association of being from a sexual minority with both tobacco and hazardous alcohol use remained stronger for women in adjusted analysis” might need to be updated as well depending on a more thorough analyses on the gender by sexual orientation interaction, as the major differences lie in the comparison between bisexual and heterogeneous individuals, and the effect of being homosexual were very similar between male and female. The discussion on why the association between tobacco use and sexual orientation attenuate can be made punchier if the authors can do a more thorough analyses based on my second point above. The statement “Differences in tobacco use between LGB and heterosexual participants in this study appear to be explained in the most part by underlying differences in major socio-demographic characteristics. This finding suggests that UK tobacco control seems to have achieved largely equitable reductions in prevalence, irrespective of sexual orientation” Is problematic. To see if tobacco control have achieved equitable reductions in prevalence between people with different sexual orientation, the authors need to use data from at least two time points to show the prevalence in tobacco use by sexual orientation. The results simply say that the association may be explained by some of the socio-demographic characteristics, and has nothing to do with whether there was an equitable reduction in prevalence.
--	---

REVIEWER	Olalekan Ayo-Yusuf
REVIEW RETURNED	Sefako Makgatho Health Sciences University (SMU), South Africa 17-Apr-2017

GENERAL COMMENTS	General comments This a generally well-written paper that uses as large representative sample to determine the association between tobacco and hazardous alcohol use and self-identified sexual orientation, while
--

controlling for socio-demographic characteristics of study participants. Using a cross-sectional study design, the study findings suggest that the high rates of tobacco use found in this population appear attributable to other socio-demographic factors while high rates for hazardous alcohol use persisted even after adjustment for other socio-demographic factors. Although further studies are needed, the paper has the potential to inform policy. Nonetheless, the authors should consider the following comments in order to improve the paper further.

Introduction

1. Pg5; lines 38-40: Could author provide a reference to support the last sentence here, if this is based on the review of the literature? If this is authors own view, then this should either be moved to the next paragraph as part justification for the second research question in this study or be placed in the discussion section as an interpretation of the authors' findings.

Methods

Sexual identity

2. Pg 7; line 33-36: Was there any reason transgender was not included as an option?

Analysis

3. Pg 9; line 25: It is well known that for outcomes with high prevalence as those determined in this study, Odds ratio from logistic regression would over estimate size of the association. Prevalence ratio on the other hand does not only provide a better estimate of association, but it also makes interpretation of results for lay reader including policy makers, easier. I would suggest that the authors consider presenting prevalence ratios from Poisson models with robust variance for example.

Discussion

4. In general, I might have missed it, but there does not seem to be any discussion in discussion section on the results on differences in the level of dependence on these substances as presented in Tables 3A and 3B (also not sure if they form part of research question 1). These findings are either discussed or the tables and related results section (see pg12; third paragraph) removed as they currently remain redundant. I nevertheless found the results interesting that bisexual women who although have a higher prevalence of tobacco use, were less dependent than the heterosexual women. Even though prevalence were found not to be independently associated with sexual identity, the issue related to the level of dependence was not explored any further e.g. What is the implication of higher use of Roll-your-own on dependence and a lower spend on tobacco among bisexuals?

5. Pg 18; lines34-36: the sentence and particularly the phrase 'even if these may not be attributable to sexual orientation alone' seem to suggest sexual orientation partly explains the disparities, which would also somewhat contradict the results from this study and the first concluding sentence of the abstract namely '...higher rates of tobacco use found among sexual minority men and women appear to be attributable to other socio-demographic factors.' Could the authors clarify?

6. Pg 19; line 25-27: to remove any confusion, at the end of the phrase '...had both lower levels of deprivation and higher smoking

	rates,' consider adding 'than heterosexuals'. 7. Pg 19; line 58: Considering that those who 'prefer-not-to-say' were less likely to be white, the lower reporting of substance use may not only be explained by societal/religious stigma, but also possible cultural stigma in this population group. In fact, this finding of this group being less likely white would support a hypothesis of cultural differences rather than religious differences currently used in the discussion section (see pg 19 line 58 and pg 20 lines 1-2). 8. Pg 20; line 25: It is suggested that the relatively small sample of subgroup of participants with a minority sexual orientation was a limitation, but the reason why is not provided. Could this lead to type II error i.e. concluding no difference when in actual fact the study was just not powered enough to detect such differences? It might be more helpful to be explicit here on the implication of such a limitation to considering/interpreting the study results, especially for tobacco use among bisexual women as compared to heterosexual women.
--	---

VERSION 1 – AUTHOR RESPONSE

Reviewer: 1

2) Strengths and limitations – second point – consider re-wording – just seemed off with the listing of goals – maybe something like “...with the aim to inform health policy, support strategic allocation of resources and, if necessary, to develop targeted interventions.

Response: In response to editor (point 1), we have now changed this section to focus solely on methodological points.

3) Introduction –Is the issue (paragraph starting ‘ there is a validated measure’... - that orientation is not reported – or the health disparities from this community are not sufficiently explored? This is a critical idea – and I would ask the authors to take a bit more space here to spell out the issues clearly before jumping to reasons why this might be the case.

Response: As we state in this paragraph, the main issue is the lack of reporting of sexual orientation in surveys also assessing health behaviours which has only started relatively recently. As a consequence, health disparities, if any, cannot be explored. We now have amended the third paragraph on page 4 to explicitly state that the lack of data is the issue. We do not doubt that there may also be other reasons which may mean that even when data are present they are not explored (e.g. political/moral/economic); however, this is beyond the remit of this paper and not relevant to the point we are trying to make here.

4) Paragraph “several possible mechanisms” – authors propose that LGB young people may be more likely to access adult venues where alcohol and cigarettes are easily available – no reference provided; England has been smoke-free for many years – is this an outdated concept (for cigarettes? I can see the argument for alcohol) -? Provide references if possible.

Response: We have now changed this sentence and provide a reference (page 5).

5) Sentence - or the use non-representative data... (the use of non-representative data?) Paragraph “There is clear need for” – consider “There is a clear need for population level studies to investigate [then state what you want to investigate].

Response: We have changed 'data' to 'sample' (i.e. use of non-representative samples would lead to biased estimates and data, resulting in spurious findings). Thank you for noting the missing 'a' which has now been included in the amended sentence.

6) Do you need to explain in the introduction why Trans folks are not included? (LGB population only?) – It is highlighted partially in the limitations (and strengths) section – but suggest being explicit.

Response: The main reason for excluding trans is that this is a study about sexual minorities and gender identity is not measured in these surveys. There may well be trans people recorded in all of the L, G, B and heterosexual groups, but we do not know as they are not recorded. However, we now explicitly clarify that this measure is developed for LGB sexual orientation minorities, given that there are other groups (e.g. asexual).

7) Methods – Table 1 – seems odd to have a table in the Methods section – but perhaps more of an editorial decision (also - no CIs for the data from ONS?)

Response: We felt it was more appropriate to present these data in the methods section to indicate reliability of the data set used compared against a national standardised data. However, we defer to the editor for a decision on where to place this table. Confidence intervals have now been added.

8) Post-16 – this might be specific to the UK? Perhaps explain?

Response: This is specific to the UK and roughly relates to education that goes beyond high school. This has been added (page 9).

9) Results – You discuss Table 2 in the results before discussing Table 1 – consider how to change this –

Response: We have now rephrased the first sentence of the results section to refer to Table 1 before Table 2.

10) Discussion – Similar findings have been revealed when dealing with other groups – I encourage the authors to include (briefly) how SES drives smoking rates/prevalence among groups based on race (for example) –

Response: We have now extended our discussion of deprivation in relation to tobacco use (page 19). However, the point we make here is that this association may in fact be different and possibly more complex with regards to sexual orientation compared with other groupings (e.g. based on race).

11) Concluding paragraph – authors presented evidence that tobacco and alcohol use were higher in the LGB community – but argued that data were not historically representative – so would caution starting the conclusion with saying these levels of use persist.

Response: We have changed 'persist' to 'exist' (page 21).

12) Adding more detail about the PNTS group – for readers not familiar with this group – perhaps in the introduction it is worth outlining why some folks might not wish to say – given that this group by numbers is larger than the other groups of interest. Further discussing how to best interpret this group in the discussion would be helpful too.

Response: We have now included a sentence on PNTS group in the introduction (page 4) and now explore further findings in relation to PNTS where there are relevant differences, i.e. in hazardous

alcohol use (page 20).

Reviewer: 2

13) First, the authors briefly reported that there was a significant interaction between gender and sexual orientation, and then stratify most of the analyses by gender. I think this is an important finding and a more proper way to follow up is to include this interaction term in the final logistic regression models. The authors can then calculate the marginal probabilities of use to see how the associations between sexual orientation and tobacco/alcohol use differ by gender. This also allows the authors to test the differences more rigorously. For example, statement like “Differences in tobacco and hazardous alcohol use as a function of sexual orientation identities were more pronounced among women than men” can be tested more rigorously by examining the interaction and follow up with tests based on marginal probabilities (or other tests similar to the simple slope analysis in linear regression). From the odds ratio estimates, it seems that the biggest differences is between male and female is the comparison between bisexual and heterosexual individual (For example, in the unadjusted hazardous drinking model, the odds ratio for bisexual for female and male were 3.75 and 1.64 respectively. But the odds ratio for homosexual for female and male were very similar – 2.53 and 2.23 respectively and there are large overlapping between the CI of these two estimates). At the moment the authors just described the differences without actually testing if the differences are statistically significant. Another advantage of this approach is that numbers of regression models run can be reduced by half, giving a more parsimonious description of the data.

Response: Thank you for this observation. We carefully considered this suggested course of action but decided against it for a number of reasons. First, the paper is already rather complex. While we agree that the inclusion of an interaction term would allow us to test the effects observed more rigorously, presenting data in this manner would be much harder to understand than stratification which is a standard approach in epidemiology in this scenario (e.g. [1]). Unstratified analysis would require twenty-eight post-hoc comparisons, as we would need to compare all levels in both men and women against each other (i.e. eight separate groups). Second, some of the covariates also show a different association with the health behaviour outcomes in men compared with women (e.g. internet access) which would be lost if a single model including an interaction term is run. Third, and most importantly, there is a theoretical rationale: sexual orientation identity and biological sex are conceptually confounded, and it would be inappropriate to combine results for this reason. For ease of interpretation and coherence, we therefore prefer to present separate models and use stratification to address the issue of effect modification. As we do provide 95% confidence intervals for prevalence, it is, in fact, possible to infer where important differences lie without a formal test. However, we have now tempered language throughout where no formal tests are conducted and provide further evidence to substantiate remarks (e.g. referring to model fit in the results section to underline the fact that sexual orientation has more of an impact on the outcomes in women than men, suggesting more pronounced differences as a function of sexual orientation identity in the former than the latter).

14) Second, the findings that the association between tobacco use and sexual orientation became non-significant after adjusting for socio-demographic factors is also very interesting. The authors have put forward a few possible explanations in the discussion. I think that the authors can potentially more accurately pinpoint which factors can account for the largest attenuation of effect. This can be done by entering the socio-demographic variables one by one into the logistic regression models.

Response: Please note that in response to point 19 of reviewer 3, we now use GLM (not logistic regression) to examine associations. However, we previously conducted stepwise (forward and backward) regression models to pinpoint which factors had the largest impact and we have retained this analysis, based on the above comment. We have now updated the results to present covariates

in order of importance (defined by variance explained in the model) in the results section and specify doing so (page 15). We now also explicitly refer to this result in the discussion section (page 18-19).

15) Some statements in the discussion are not accurate. For example, "In general, it appears that differences were more pronounced among women than men, as has been previously reported". It seems that the only gender differences were in the comparison between bisexual and heterogeneous individuals, but the gender differences in the comparison between homogenous and heterogeneous individuals were similar. This can be tested using the authors' data and needs to be clarified in the discussion. The statement "Moreover, the importance of targeting health messages to lesbian/bisexual women differently to gay/bisexual men is underlined by the association of being from a sexual minority with both tobacco and hazardous alcohol use remained stronger for women in adjusted analysis" might need to be updated as well depending on a more thorough analyses on the gender by sexual orientation interaction, as the major differences lie in the comparison between bisexual and heterogeneous individuals, and the effect of being homosexual were very similar between male and female.

Response: We assume the reviewer means heterosexual rather than heterogenous here? Our response to point 13 is relevant here, including our argument that 95% confidence intervals for prevalence allow inference about where important differences lie. Insofar that this argument is accepted, then we prefer to retain the summary that differences appear greater among women. We have deleted 'in general' and note that our language is 'it appears' and is supported by previous findings. With regards to the second point, the amended analysis (in response to point 19 of reviewer 3), has produced supportive results given that differences in hazardous alcohol use persist in sexual minority women but not men after controlling for various confounders.

16) The discussion on why the association between tobacco use and sexual orientation attenuate can be made punchier if the authors can do a more thorough analyses based on my second point above. The statement "Differences in tobacco use between LGB and heterosexual participants in this study appear to be explained in the most part by underlying differences in major socio-demographic characteristics. This finding suggests that UK tobacco control seems to have achieved largely equitable reductions in prevalence, irrespective of sexual orientation" Is problematic. To see if tobacco control have achieved equitable reductions in prevalence between people with different sexual orientation, the authors need to use data from at least two time points to show the prevalence in tobacco use by sexual orientation. The results simply say that the association may be explained by some of the socio-demographic characteristics, and has nothing to do with whether there was an equitable reduction in prevalence.

Response: Regarding the first point, please see response to point 14 above. Regarding, the second point, we agree with the reviewer and have now removed this statement from the discussion.

Reviewer: 3

17) Introduction Pg5; lines 38-40: Could author provide a reference to support the last sentence here, if this is based on the review of the literature? If this is authors own view, then this should either be moved to the next paragraph as part justification for the second research question in this study or be placed in the discussion section as an interpretation of the authors' findings.

Response: Thank you. We have now moved this sentence into the next paragraph as justification for the research question (page 9).

18) Methods Pg 7; line 33-36: Sexual identity - Was there any reason transgender was not included as an option?

Response: The sexual identity measure is based on a validated measure of sexual orientation identity that is used in the UK as described in the methods section (see also response to point 6 by reviewer 1) and trans is not a sexual identity but a gender identity issue.

19) Analysis Pg 9; line 25: It is well known that for outcomes with high prevalence as those determined in this study, Odds ratio from logistic regression would over estimate size of the association. Prevalence ratio on the other hand does not only provide a better estimate of association, but it also makes interpretation of results for lay reader including policy makers, easier. I would suggest that the authors consider presenting prevalence ratios from Poisson models with robust variance for example.

Response: Thank you for this comment. We agree and have now changed the analysis to calculate prevalence (risk) ratios.

20) Discussion: In general, I might have missed it, but there does not seem to be any discussion in discussion section on the results on differences in the level of dependence on these substances as presented in Tables 3A and 3B (also not sure if they form part of research question 1). These findings are either discussed or the tables and related results section (see pg12; third paragraph) removed as they currently remain redundant. I nevertheless found the results interesting that bisexual women who although have a higher prevalence of tobacco use, were less dependent than the heterosexual women. Even though prevalence were found not to be independently associated with sexual identity, the issue related to the level of dependence was not explored any further e.g. What is the implication of higher use of Roll-your-own on dependence and a lower spend on tobacco among bisexuals?

Response Thank you for this comment. We now have changed question 1 to make explicit that this formed part of the question and refer to the findings in relation to use characteristics in the discussion (pages 19-20).

21) Pg 18; lines34-36: the sentence and particularly the phrase 'even if these may not be attributable to sexual orientation alone' seem to suggest sexual orientation partly explains the disparities, which would also somewhat contradict the results from this study and the first concluding sentence of the abstract namely '...higher rates of tobacco use found among sexual minority men and women appear to be attributable to other socio-demographic factors.' Could the authors clarify?

Response: What we mean to say here is that while general socio-demographic characteristics may be the main driver for disparities in tobacco use (rather than factors which may be specifically related to sexual orientation), these disparities still exist (as shown by unadjusted analyses) and therefore cannot be ignored. The sentence has now been rephrased to make it consistent with the rest of the paper (page 18).

22) Pg 19; line 25-27: to remove any confusion, at the end of the phrase '...had both lower levels of deprivation and higher smoking rates,' consider adding 'than heterosexuals'.

Response: This has been done (please note that this paragraph has been restructured so the sentence now appears later on page 19).

23) Pg 19; line 58: Considering that those who 'prefer-not-to-say' were less likely to be white, the lower reporting of substance use may not only be explained by societal/religious stigma, but also possible cultural stigma in this population group. In fact, this finding of this group being less likely white would support a hypothesis of cultural differences rather than religious differences currently used in the discussion section (see pg 19 line 58 and pg 20 lines 1-2).

Response: Thank you for this observation. We have now added this explanation to this section (page 20).

24) Pg 20; line 25: It is suggested that the relatively small sample of subgroup of participants with a minority sexual orientation was a limitation, but the reason why is not provided. Could this lead to type II error i.e. concluding no difference when in actual fact the study was just not powered enough to detect such differences? It might be more helpful to be explicit here on the implication of such a limitation to considering/interpreting the study results, especially for tobacco use among bisexual women as compared to heterosexual women.

Response: We now explicitly mention that power may have been a problem (page 20). However, we should also note that the prevalence ratio estimates which - unlike the confidence intervals - are not affected by power showed a clear attenuation, so that differences missed, if any, were likely rather small.

References

1. Rothman KJ, Greenland S, Lash TL. Modern Epidemiology, 3rd Ed. Philadelphia, USA: Lippincott Williams & Willkins; 2008.

VERSION 2 – REVIEW

REVIEWER	Ryan Kennedy Johns Hopkins Bloomberg School of Public Health, USA
REVIEW RETURNED	19-Jun-2017

GENERAL COMMENTS	This is my second look at this paper - the authors responded well/adequately to my concerns - and those of other reviewers.
---

REVIEWER	Olalekan Ayo-Yusuf Sefako Makgatho Health Sciences University, Medunsa. South Africa
REVIEW RETURNED	11-Jun-2017

GENERAL COMMENTS	No additional comments.
-------------------------